# Characterization of Key Aroma-Active Compounds Isolated from Omija Fruit Treated Differently Based on Odor Activity Values and Descriptive Sensory Analysis

**DOI:** 10.3390/foods9050638

**Published:** 2020-05-15

**Authors:** Mina K. Kim, Hae won Jang, Kwang-Geun Lee

**Affiliations:** 1Department of Food Science and Human Nutrition and Fermented Food Research Center, Jeonbuk National University, 567 Baekjedaero, Deokjin-gu, Jeonju-si, Jeonbuk 54896, Korea; minakim@jbnu.ac.kr; 2Korea Food Research Institute, 245, Nongsaengmyeong-ro, Iseo-myeon, Wanju-gun, Jeollabuk-do 55365, Korea; hwjkfri@kfri.re.kr; 3Department of Food Science and Biotechnology, Dongguk University-Seoul, 32, Dongguk-ro, Ilsandong-gu, Goyang-si, Gyeonggi-do 10326, Korea

**Keywords:** omija, GC-MS, sensory threshold, flavor analysis, descriptive analysis, odor activity value

## Abstract

The objective of this study was to characterize the key aroma-active compounds isolated from omija fruits using gas chromatography-mass spectrometry (GC-MS), orthonasal threshold testing, and descriptive sensory analysis techniques. Omija fruits were prepared using four different methods: raw, pureed, freeze-dried, and hot-air dried. The aromatic compounds were extracted with ethanol. Flavor compounds were further isolated using distillation under reduced pressure, followed by liquid–liquid extraction, and were then identified using GC-MS. A total of 40 volatile compounds were identified in omija fruits; nine were further identified as key aroma-active compounds found in omija fruits. The odor-active values for key flavor compounds were calculated, and aroma descriptions perceived by humans were determined using a highly trained panel. This study found that compounds high in omija fruit extracts were not necessarily the odor active compounds and 4-terpineol (1.84) and *α*-terpineol (2.58) were high odor-active compounds in omija fruits. Samples with high levels of the two compounds (hot air- and freeze-dried omija fruit) had high intensities of “spicy” and “wet-wood” aromatics.

## 1. Introduction

Omija (*Schisandra chinesis Baillon*) originates from East Asia in countries including Korea, Japan, Russia, and China, where it is also harvested. The literal meaning of omija in Korean is “berries with five distinctive flavors”. These flavors include sweet, salty, sour, bitter, and astringent [1]. Omija has very distinctive sensory characteristics. Due to these, many studies have been conducted to develop value-added food products with different levels of added omija such as demi-glace sauces [2], Kimchi [3,4], omija cookies [5], omija wines [6], and makgeolli with omija extracts [7], all utilizing omija extracts and/or omija fruits. These studies proved the potential for omija fruits in ingredient application, however, to expand its scope of application in different food categories, it is necessary to understand the flavor characteristics of omija.

To date, only limited research has been conducted to characterize the flavor profiles of omija fruits. Previous study has reported the key aroma-active compounds of omija fruits cultivated in different regions of Korea—northeast, mideast mountain area, and the southwest region—using a gas chromatography (GC)-mass selective detector (MSD) and aroma extraction dilution analysis (AEDA), followed by GC-olfactometry (GC-O). Their work reported finding terpene hydrocarbons including *α*- and *β*-ylangene, *α*- and *β*-selinene, (*E*)-*β*-farnesene, and germacrene D as the major volatile aromatic compounds associated with omija fruits, regardless of the cultivating region in Korea [1]. Another study reported the volatile flavor profiles of omija fruits [8]. Their work utilized a steam distillation-extraction method (SDE) prior to GC-MS, and reported finding sesquiterpenes (including *α*-ylangene, *β*-elemene, *β*-cibebene, aristolene, and *γ*-selinene), monoterpenes (such as mycene and *γ*-terpinene), and terpene alcohols (including T-murolol and terpinen-4-ol) from dried omija fruits [8].

The instrumental volatile flavor analysis only provided the quantity of qualified compounds; therefore, the roles of identified volatile compounds in sensory perception, and their potential interactions with other compounds and the food matrix, are still unknown. To understand the roles of these volatile aroma compounds in the complex food matrix, the odor activity of each volatile flavor compound should be determined. The odor activity values (OAVs) are the ratios of concentration found in the food matrix that meet the sensory orthonasal threshold for volatile aromatic compounds [9]. Previous studies on Cheddar cheese utilized OAVs for identifying key aroma-active compounds responsible for reduced- and full-fat Cheddar cheese using cheese model studies [9]. Their study demonstrated the different OAVs of volatile compounds according to polar- and/or non-polar matrices (reduced- and/or full-fat Cheddar cheese model). The OAVs were determined in different food categories including Parmigiano Reggiano cheeses [10], strawberries [11], Jiashi muskmelon juice [12], and durian wines [13].

To our knowledge, neither the volatile flavor analysis nor determinations of odor activity of each volatile flavor compound identified in omija fruits have been comprehensively studied. Therefore, the objective of this study was to characterize the key aroma-active compounds isolated from omija fruits using GC-MS, orthonasal threshold testing, and descriptive sensory analysis techniques.

## 2. Materials and Methods

### 2.1. Omija Extract Samples and Chemical Reagents

Omija fruits were obtained from a local producer (Hyojongwon^®^) located in Munkyung, Gyeongsangbukdo in Korea. Munkyung is one of three locations in Korea famous for omija cultivation [1]. Upon receiving the omija fruits, they were stored in a deep freezer at −80 °C until analysis. Due to the perishability and fragility, omija fruits are distributed either frozen or dried in Korea. Therefore, omija extracts were prepared from four different scenarios of omija distribution condition: raw omija fruit (non-frozen), pureed omija fruit (raw omija fruits blended), dried omija fruit, and freeze-dried omija fruits. All were extracted in the same manner using 35% (*v*/*v*) ethanol. Prior to ethanol extraction, the pureed omija sample was prepared by grinding frozen omija fruits using a food processor (SMX-3500GN, Shinil, Seoul, Korea). The dried omija was prepared by drying omija fruits at 60 °C for 12 h. The freeze-dried omija was prepared by freeze-drying at −72 °C for 48 h with the pressure set at 6.67 Pa using a freezer-dryer (model #FD-8508, ILSHIN Lab, Seoul, Korea). The freeze-dried omija was then ground into a particle size of 710 µm for ethanol extraction. Once all the omija samples—raw, pureed, dried, and freeze-dried—were prepared, active ingredients from the omija fruits were extracted using food-grade ethanol (35% *w*/*v*). Flavor extraction condition was optimized from the authors’ preliminary studies. Five grams of omija fruit was added to 100 mL of ethanol and kept in a water bath shaker for seven days. Once extraction was completed, the ethanol was removed using a rotary evaporator (model R-3 HB, Buchi, Flawil, Switzerland) under reduced pressure at 50 °C with 90 mmHg (0.012 Mpa). All chemical reagents used in the study were purchased from Sigma-Aldrich Corporation (St. Louis, MO, USA).

A liquid–liquid continuous extraction (LLE) was conducted on the ethanol-extracted omija samples. Dichloromethane (DCM) was used to extract the volatile aromatic compounds from the samples with apparatus for continuous extraction. The aromatic compounds were removed from the extract to DCM for 6 h. After this, 20 g of anhydrous sodium sulfate was added to DCM to remove the moisture and kept for 24 h. Then, then DCM was evaporated for 15 min using the rotary evaporator to a volume of about 1 mL, and further evaporated to a volume of 0.4 mL with liquid nitrogen for 10 min.

### 2.2. Gas Chromatography-Mass Spectrometry (GC-MS)

The extracted volatiles were analyzed using a 7820A gas chromatograph (GC) system coupled with a 5977E mass selective detector (Agilent Technologies, Palo Alto, CA, USA). The gas chromatograph was equipped with a 60-m DB-WAX column (internal diameter = 0.32 mm, thickness = 0.5 μm; J&W Scientific, Folsom, CA, USA). The inlet temperature of the GC was 200 °C, and the oven temperature was maintained at 150 °C for 2 min, then programed to increase to 200 °C at 3 °C/min, and held for 30 min. Helium was used as a carrier gas with a constant flow rate of 1.2 mL/min. The transfer line, ion source, and quadrupole were maintained at 250 °C, 230 °C, and 150 °C, respectively. The mass spectra were employed in full scan mode, and the mass range was collected between *m*/*z* 35 to 400. The tentative identification of volatile compounds was determined by a comparison of the retention indices and mass spectra with those of the mass spectral library and co-injection with authentic compounds in the authors’ laboratory (Wiley and NIST 08). The quantification of identified volatile compounds was calculated by the peak area ratio (peak area of each compound/peak area of the internal standard: 100 µL of 10 g/L of phenethyl alcohol). All analysis was conducted in triplicate.

The mass of each compound was analyzed using an analytical balance [14]. The percentage of total peak area of volatile components was determined by GC-MS. The total mass of the volatile component was calculated by multiplying the mass of the concentrate by total peak area percentage of the volatile component. The mass of each volatile compound was obtained by multiplying the total mass calculated above with the percentage of each component.

Concentration of each volatile compound (μg/g)
(1)$$=\frac{\mathrm{weight}\;\mathrm{of}\;\mathrm{condensed}\;\mathrm{extract}\times \mathrm{GC}\;\mathrm{peak}\;\mathrm{area}\;\%}{\mathrm{weight}\;\mathrm{of}\;Schisandra\;chinensis\times 100}\times 10^{6}$$

### 2.3. Orthonasal Threshold Test

Orthonasal threshold testing was conducted in accordance with the American Society for Testing and Materials procedure E679-9, an Ascending Forced-Choice (AFC) method of limits, as stated in previous study [9]. Deodorized water was used as a diluting media, and prepared by boiling distilled water until two-thirds of the original volume remained. A stock solution of selected volatile aromatic compounds was prepared in 75% ethanol, and aliquots of stock solutions were serially diluted (factor of three; 3AFC method) and added to the deodorized water. These diluted solutions were poured into 40 mL white plastic cups (Cleanwrap, Gimhae, Korea). In each series, blank samples were prepared with deodorized water. All cups were labeled with a three-digit code. All samples and stock solutions for orthonasal threshold testing were prepared within a day of testing, and were poured into cups 3 h before the test to achieve headspace equilibrium in the cup [9,15].

Prior to testing, all participants were instructed on the threshold test protocol, and asked to select one sample from a row of three samples, repeating the procedure seven times (7-3AFC method of limit). The individual best estimate thresholds (BETs) were calculated as the geometric mean of the last concentration with an incorrect response and the first concentration with a constant correct response. Use of sure/not sure adjustment was applied, and the concentration was increased by a factor of 1.41 if the participant was “not sure” about their choice [9,16]. This sure/not sure adjustment minimizes the errors from a chance correct response. Once individual BETs were calculated, a group BET value was calculated as the geometric mean of individual BET values. A total of 40 subjects participated in the orthonasal threshold testing for each compound.

### 2.4. Odor Activity Value (OAV) Calculation

The OAVs for each compound were calculated from the concentration of the compound found in the omija extract divided by the BET value, since OAV is the ratio of compound concentration in the food matrix to the sensory odor threshold for the compound [9,17].

### 2.5. Descriptive Sensory Analysis

Descriptive analysis was conducted on four omija samples, and only the flavor components were analyzed. A 25 mL sample of the ethanol-extracted omija fruits was prepared in a 125 mL brown amber glass jar, and prepared 2 h before the evaluation in order to reach the headspace equilibrium. The brown amber glass jar (Duran^®^, Louisville, KY, USA) used in this study was put in an oven at 200 °C for 12 h and cooled to room temperature before adding the omija samples to minimize the residual volatile aroma compounds in the glass jar.

A highly trained panel consisting of five panelists (four females and one male, aged 23 to 39) evaluated the aroma components of the omija fruits. Each panelist had received more than 250 h of training and evaluation experience with various food products including soybean processed foods, omija extracts, fruit-flavored beverages, and dairy beverages using the Spectrum™ method. Panelists had 2 h calibration sessions with a basic taste solution. A sensory lexicon was generated after each calibration session, and a definition and reference for each sensory attribute was also generated to minimize panelist variations for describing the sensory terms. During the actual taste session, omija samples were prepared in a brown jar, labeled with random three-digit numbers, and each panelist was asked to evaluate the aroma of each sample in triplicate. The panelists entered their responses in a paper ballot using a 15-pt universal scale in the Spectrum™ method. A 2-min rest was enforced between samples to minimize any carry-over effect.

### 2.6. Statistical Analysis

Results are presented as averages ± standard deviation (SD) from triplicate measurements of the analyses. Analysis of variance (ANOVA) followed by Fisher’s Least significant Difference (LSD) test were conducted at the *α* = 0.05 level. Principal component analysis (PCA) was conducted to determine where the omija samples were located in the volatile flavor and sensory characteristics map, and was carried out in XLSTAT (v.2017, Addinsoft, Paris, France). 

## 3. Results

### 3.1. Instrumental Flavor Analysis Result

A total of 26 volatile flavor compounds were identified from the omija fruit extracts (Table 1). When looking at the effect of drying methods on the volatile flavor compound profiles of omija extracts, 18 compounds were identified in the raw omija fruit extract and the hot-air dried omija fruit extract and freeze-dried omija fruit extracts, respectively, while there were 11 compounds in the pureed omija fruit extract. When comparing volatile aromatic compounds between the four omija fruit extract samples, 10 compounds matched from the pureed omija fruit extract, 13 matched from freeze-dried omija fruit extract, and 14 compounds matched to the raw omija fruit extract matched those identified in the raw omija fruit extract samples. Six compounds—1,8-cineol, cyclohexenol, 4-terpineol, *α*-terpineol, *β*-citronellol, and farnesol—were found in all four omija fruit extracts.

A previous study reported that germacrene D, *α*-ylangene, *α*-selinene, *β*-selinene, *β*-elemene, and (*E*)-*β*-farnesene were the major volatile flavor compounds in omija fruits cultivated in Korea [1]. Only three terpene hydrocarbons—*α*-selinene, *β*-selinene, and ylangene—were found in this study, which matches those identified in a previous study [1]. For example, *α*- and *β*-selinene were detected in all omija fruit extracts except for the pureed extract, while ylangenes were only detected in the freeze-dried extract. Among the volatile compounds identified in the previous study, five compounds—(*E*)-*β*-ocimene, *α*-pinene, hexanal, *α*-terpinene, and 5-methyl furfural—were identified as aroma-active compounds from dried omija fruits using GC-olfactometry with aroma extract dilution analysis (AEDA). Among them, only (*E*)-*β*-ocimene was detected from the raw and hot-air dried omija fruit extracts in this study. Other aroma-active compounds previously reported were not detected in this study [1]. A recent study reported volatile compounds from omija fruits using Stir-bar sorptive extraction followed by GC-MS, and reported 28 major volatile flavor compounds found in omija fruits treated differently: frozen, frozen-blended, and freeze-dried [18]. Among the 28 volatile flavor compounds, nine compounds matched the flavor compounds identified in this study: *β*-myrcene, 1,8-cineol, *γ*-terpinene, *β*-ocimene, 4-cymene, undecanone, 4-terpineol, *α*-copaene, and ylangene. As with the other study, freeze-dried omija fruits had the most abundant aroma profiles.

### 3.2. Orthonasal Threshold and Odor Activity Values of Key Flavor Compounds

The BET values of nine key volatile compounds from omija fruit extracts as well as odor activity values (OAVs) can be found in Table 2. These nine key volatile compounds were selected based on the volatile flavor analysis results collected in this study as well as in a comparison to the previously published data so that the volatile flavor compounds with high peak area ratio in this study were cross-checked with previously reported odor-active compounds [1,18]. Based on this process, nine key compounds were selected. The BET values of the nine volatile compounds were within the range of previously published BETs [19,20,21]. For example, the BET for *β*-myrcene in this study was 9.46 µg/kg, while the previously reported BET was 13–15 µg/kg in water; the BET for 1,8-cineole in this study was 1.86 µg/kg, while the previously reported BET was 12 µg/kg [22]. Similarly only 10-fold differences were observed in bornyl acetate (13.13 µg/kg) and undecanone (98.5 µg/kg) when compared to the reported BET values of 75 µg/kg and 7 µg/kg, respectively. Considering that individual and group BETs can vary up to 1000 times [15], the BETs collected in this study matched the ranges of previously reported BET values.

OAVs were calculated using the BETs from this study. An OAV is one method of estimating the contribution of volatile compounds to flavor because OAV considers both the sensory thresholds and the concentrations in the food matrix [9]. From the raw omija fruit extract, *α*-terpineol and 4-terpineol showed the highest OAVs, with 1.51 and 1.13, respectively. Similarly, these two compounds showed high OAVs in all the other omija fruit extracts. For example, the OAVs for 4-terpineol were 1.46, 1.84, and 2.16 for the pureed, freeze-dried, and hot-air dried omija fruit extracts, respectively. Similarly, the OAVs for *α*-terpineol were 1.63, 2.58, and 2.16 for the pureed, freeze-dried, and hot-air dried omija fruit extracts, respectively. The BET values of two compounds—4-terpineol and *α*-terpineol—were 3.31 and 1.00 µg/kg, which were relatively lower than for other compounds. High OAVs of 4-terpineol and *α*-terpineol in omija fruit extracts can be due to low BET values. In previous studies, *α*-terpineol was reported as a major aroma constituent of red and green huajiao (*Zanthoxylum bungeanum* and *Zanthoxylum schinifolium*) [23]. To our knowledge, this is the first study to report that 4-terpineol and *α*-terpineol are high odor-active compounds in omija fruits.

### 3.3. Descriptive Sensory Analysis Result

With OAV information on each key volatile compound, sensory analysis using a highly trained panel was conducted to understand the impact of each compound on human sensory perception. Descriptive sensory analysis results including aroma attribute (aroma term), definition, and reference, and the mean values for each sensory attribute can be found in Table 3. Only the aroma attributes were evaluated in this study because omija extracts, instead of foods with added omija extracts, were evaluated.

A significant difference in the overall aroma intensity was observed among the four samples: the freeze-dried and hot air-dried omija extracts had values of 2.83 and 2.50, respectively, on a 15-pt. universal intensity rating scale, which were significantly higher than the intensities found in the raw and pureed omija extract samples (*p* < 0.05). Drying processes are widely utilized in the food industry to avoid biological degradation and to reduce shipping and storage costs. However, these processes are known to affect flavor retention [24]. Freeze-drying keeps the aromatic profiles of raw products with minimal deterioration [25,26]. Higher intensities of the overall aroma impact of the two dried omija samples (hot-air dried and freeze-dried) may be attributed to the greater contact area with the solvent during the ethanol extraction process. From the instrumental flavor analysis results (Table 1), it was observed that the peak area under the curve captured for the two dried omija samples (hot air-dried and freeze-dried) were higher than for the other two samples. A similar pattern was also observed among the findings from the trained human assessors, in that higher aroma intensities were observed in dried omija extract samples (freeze- and hot air-dried).

The descriptive sensory analysis utilized sensory lexicon to describe the sensory characteristics of the food products for objective profiling. These sensory lexicons had been previously developed for different product categories including [27], Cheddar cheese [28], rooibos tea [29], orange juice [30], and pomegranate juice [31]. This study developed a sensory lexicon for omija fruit extracts, and seven aroma terms were developed. Among them, significant differences of “spicy,” “smoky,” “wet wood,” “tomato ketchup,” and “alcohol” aromatics were observed (*p* < 0.05). A spicy aromatic in this study was defined as being similar to the characteristic aromatics associated with whole black pepper. A previous study on the sensory analysis of omija fruits, and not omija extracts, reported a “ginger” attribute to describe the aroma characteristics of omija fruits [18]. According to another previous study, “ginger”, “black pepper”, and “red pepper” attributes share similar sensory sensations as bitter, citrus, heat, musty, and soapy characteristics [32]. The spicy aromatics were significantly higher in the hot air-dried omija extract samples (1.90) for the dried omija extract, and 1.43 for the freeze-dried omija extract than the raw and pureed omija extracts (*p* < 0.05). This could be due to the active ingredients responsible for spicy aromatic characteristics being extracted more efficiently from the dried omija extract samples than the others, possibly due to the lack of moisture in the omija fruit during the ethanol extraction process. This spicy aromatic showed a high correlation to overall aroma intensities (*r* = 0.986, data not shown). Smoky aromatics were significantly higher in the dried omija extracts (freeze- and hot-air dried) than in the other extract samples (*p* < 0.05). The hot air-dried omija extracts were prepared by drying the omija fruits at 60 °C for 12 h, and this mild heat may have influenced the smoky aromatics associated with this sample.

In order to correlate the key volatile compounds with known OAVs to sensory characteristics of omija fruits prepared with different methods (raw, pureed, freeze-dried and hot-air dried), principal component analysis (PCA) was conducted (Figure 1a,b). The PCA biplot represents the orthogonal transformation of variable data to show the impact of relevant variables (in this study, instrumental volatile flavor analysis and sensory analysis results); therefore, the association between variables can be easily seen in the PCA biplot. In this study, two PCA biplots were included: Figure 1a represents the transformed data in PC1 and PC2, which explains 85.42% of total data variability, and Figure 1b includes the data variables in PC1 and PC3, which explains a total of 68.77% of data variability. A PCA biplot with PC1 and PC2 explains 85.4% of data, therefore 14.58% of total variability in this study is still not explained by Figure 1a; therefore, Figure 1b was included to maximize the data explanation. Hot-air dried omija samples had dissimilarity in sensory and instrumental flavor characteristics in comparison to the other samples in both PCA biplots (Figure 1a,b). Raw omija, freeze-dried omija, and pureed omija extracts were located in the left of PC1 and seemed to share similar characteristics in the PCA biplot with PC1 and PC2 (Figure 1a). When looking at the location of these samples in the PCA biplot with PC1 and PC3 (Figure 1b), the raw and pureed omija extract samples shared similarity, and the freeze-dried omija extract sample was placed at a distanced location in Figure 1b. The PCA biplot is a multidimensional technique, in that different dimensions selected for the PCA biplot can demonstrate different locations of variables. Regardless, the hot air-dried omija sample showed dissimilarity to other samples based on two PCA biplots. For PCA analysis, nine volatile compounds selected for OAV calculations in this study (Table 2) were included as data variables in PCA because these compounds have been previously well-documented as odor active compounds based on volatile flavor analysis [1,18].

Hot-air dried omija was characterized with high intensities of 1,8-cineol, *γ*-terpinene and *α*-, and 4-terpineol compounds with sensory characteristics of cough syrup, spicy, and alcohol. 1,8-cineol, also known as eucalyptol, has spice-like aromatics [18,20]. Looking closely into the association between volatile aromatic compounds to sensory perception, two PCA biplots (Figure 1a,b) were considered together. 1,8-cineol was highly associated with alcohol aromatics in both PCA biplots (Figure 1a,b) and the correlation analysis results also confirmed a strong correlation between 1,8-cineol and alcohol aromatics (*r*^2^ = 0.980; data not shown). A high correlation between *γ*-terpinene and cough syrup was also noted in both PCA biplots (Figure 1a,b), and correlation analysis confirmed a strong correlation (*r*^2^ = 0.992). In the case of *α*-, and 4-terpineol, a high correlation of these two compounds to spicy aromatics was noted in the PCA with PC1 and PC2 (Figure 1a), while only *α*-terpineol showed a high association to spicy aromatic in PCA with PC1 and PC3 (Figure 1b). In other sword, the angle between *α*-, and 4-terpineol was small in the PCA biplot (Figure 1a) with PC1 and PC2, indicating the strong association between two compounds. However, the PCA biplot with PC1 and PC3 (Figure 1b) showed that the angle between the two compounds was slightly bigger, and confirms that *α*-terpeneol is more strongly associated with spicy aromatics than 4-terpineol. The correlation analysis results in this study confirmed that *α*-terpineol is more correlated to spicy aromatics (*r*^2^ = 0.994; data not shown). In a previous study, terpene compounds such as terpinene and terpineol have been shown to provide spice-like and herbal aromatics [1]. The above-mentioned aroma characteristics were abundantly found in hot-air dried omija fruit. It is worth noting that not all compounds showed a high association with hot air-dried samples such as 1,8-cineol, *γ*-terpinene, *α*-, and 4-terpineol, were the key aroma active compounds in hot-air dried omija fruit sample. For example, the OAVs of 1,8-cineol and *γ*-terpinene in hot air-dried omija were low (0.20, and 0.00, respectively; Table 2), meaning that these two compounds were low aroma impact compounds in the hot air-dried omija sample. *α*-, and 4-terpineol, however, were key aroma impact compounds in hot air dried omija as these compounds had high OAV values (1.13 and 2.16, respectively, Table 2).

A higher correlation of undecanone and *β*-ocimene to tomato ketchup and lemon-like aromatics, and these were characteristic aromatics of raw omija fruits in the PCA biplot with PC1 and PC2 (Figure 1a). When looking at different data space, undecanone was more correlated to tomato ketchup-like aromatics (*r*^2^ = 0.930; data not shown) and were the characteristic aromatics in the raw omija extract sample (Figure 1b), while *β*-ocimene was more correlated to lemon-like aromatics (*r*^2^ = 0.711; data not shown), which were the characteristic aromatics in the freeze-dried omija extract based on the PCA biplot with PC1 and PC3 (Figure 1b). Undecanone were previously reported to produce “fruity” and/or “fecal” aromatics based on its concentration [33] and were previously identified as abundantly found volatile compounds in frozen omija fruit samples [18]. While undecanone was highly associated with the raw omija extract, the OAV of undecanone in the raw omija fruit extract was 0.00. This is another case where this compound may not be the odor-active compound in raw omija fruit, while the concentration found in the raw omija extract was high. High correlation between *β*-ocimene and *β*-citronellol was also noted in both PCA biplots with *r*^2^ = 0.830, as the angle between the two compounds was small in both PCA biplots (Figure 1a,b). Both compounds have been previously reported to produce “rose-floral fragrances in tree peony cultivars” [34]. High correlation between *β*-myrcene and smoky and wet wood aromatics were noted in the PCA biplot with PC1 and PC2 (Figure 1a), which is in agreement with the previously reported aroma description of *β*-myrcene as pulp, and woody with citrus and sweet notes [35].

## 4. Conclusions

This study used four different omija fruit samples: raw, pureed, freeze-dried, and hot-air dried, and the flavor components were ethanol extracted. Then, odor-active values for key flavor compounds collected from the omija fruit samples were calculated, and aroma profiles perceived by humans were determined using a highly trained panel. Hot air-dried omija fruits had the most versatile flavor profiles than other omija fruits treated differently (raw omija, pureed omija, and hot air-dried omija fruit) based on the trained sensory panel evaluation. This study found that the high compounds in omija fruit extracts were not necessarily the odor active compounds. Among the volatile compounds identified in omija extracts prepared differently, 4-terpineol and *α*-terpineol were identified as high odor-active compounds in omija fruits with high OAVs. Samples with high levels of these two compounds (hot air and freeze-dried omija fruit extracts) had high intensities of “spicy” and “wet-wood” aromatics.

## Figures and Tables

**Figure 1 foods-09-00638-f001:**
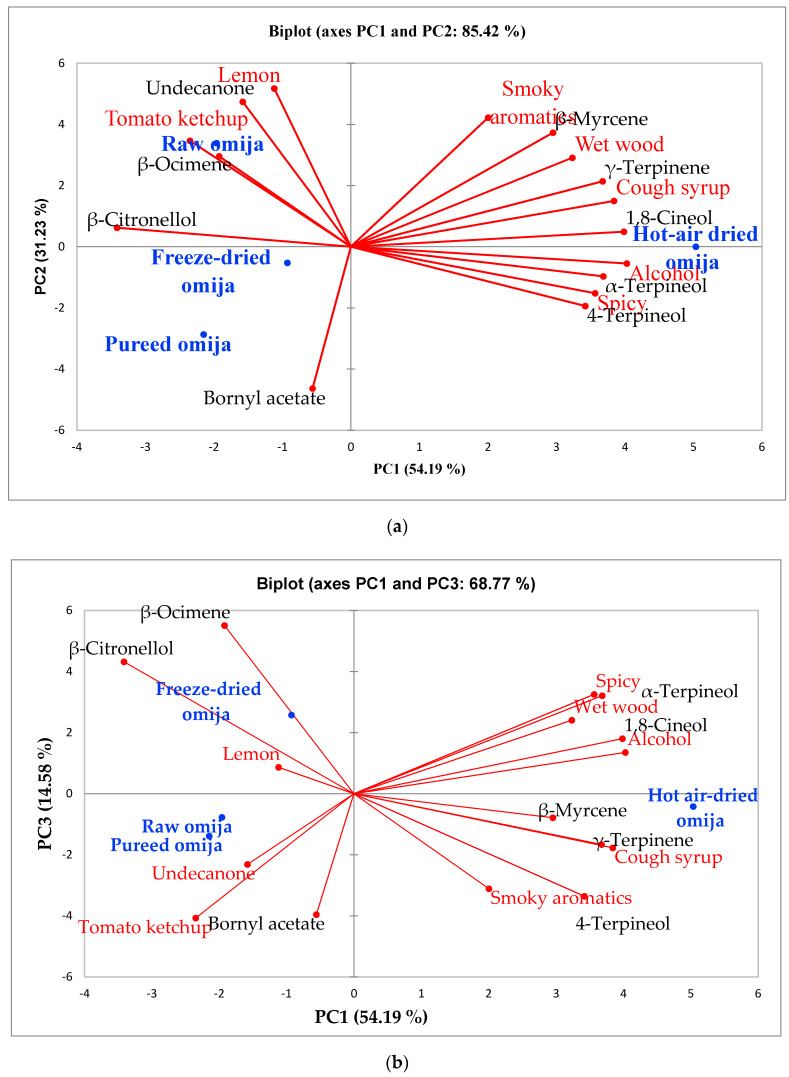
(**a**) Principal component bioplot of sensory and instrumental flavor analyses on four omija samples with PC1 and PC2. (**b**) Principal component bioplot of sensory and instrumental flavor analyses on four omija samples with PC1 and PC3.

**Table 1 foods-09-00638-t001:** Instrumental flavor analysis results of omija fruit extracts.

Peak No.	Compound		Kovats Index ^1^		mg/kg			
		ID Method	KI	KI (Ref)	Raw Omija Fruit	Pureed Omija Fruit	Freeze Dried Omija Fruits	Hot Air-Dried Omija Fruits
1	*β*-Myrcene	MS, Co, KI	1177	1172	1.88		2.5	0.74
2	1,8-Cineol	MS, Co, KI	1232	1224	0.22	0.13	0.71	0.37
3	*γ*-Terpinene	MS, Co, KI	1264	1261	0.63		1.49	0.14
4	*β*-Ocimene	MS, Co		1245	0.84			1.03
6	4-Cymene	MS, Co, KI	1289	1287				0.76
7	Cyclohexenone	MS			0.37	0.67		
8	Cyclohexenol	MS			0.27	0.74	1.15	0.22
9	Bornyl acetate	MS, Co, KI	1611	1613	0.54	2.51	1.19	0.81
10	Undecanone	MS, Co, KI	1619	1615	0.34			
11	4-Terpineol	MS, Co, KI	1631	1633	3.74	4.82	6.08	3.73
12	Dimethyloctadiene	MS			1.00		3.42	
13	Citronellylpropianoate	MS						2.42
14	*γ*-Selinene	MS, Co, KI	1724	1756	0.77		4.07	5.87
15	α-Terpineol	MS, Co, KI	1586	1576	1.51	1.63	2.58	2.16
16	*β*-Selinene	MS			0.91	5.82	0.23	2.26
17	α-Selinene	MS			0.55		3.4	3.16
18	*β*-Citronellol	MS, Co, KI	1783	1774	1.41	1.07		1.9
19	*β*-Farnesene	MS, Co, KI	1683	1674	1.76	0.82	1.23	0.94
20	*β*-Verbesinol	MS					0.23	
21	*γ*-Gurjunene	MS					0.27	
22	α-Copaene	MS, KI	1515	1509		1.19		
23	Bicyclosesqui-phellandrene	MS, KI	1795	1782			0.86	1.12
24	Ylangene	MS, Co, KI	1515	1499			0.2	
25	α-Cadinol	MS, Co, KI	2264	2264	3.85	4.22	2.17	0.77
26	11-Selinenol	MS, KI			5.02		0.21	2.3

^1)^ Identification methods of volatile compounds compared with mass spectrum (MS) in Wiley mass database, compared to the Kovat Index (KI) on a DB-wax column and co-injection (CO) with authentic chemicals.

**Table 2 foods-09-00638-t002:** Best estimate threshold (BET) values and odor activity values (OAVs) for key aroma compounds.

Peak No.	Compound	BET (µg/kg)	OAV			
			Raw Omija Fruit Extract	Pureed Omija Fruit Extract	Freeze Dried Omija Fruit Extract	Hot Air-Dried Omija fruit Extract
1	*β*-Myrcene	9.46	0.20	0.00	0.26	0.08
2	1,8-Cineol	1.86	0.12	0.07	0.38	0.20
3	*γ*-Terpinene	389.40	0.00	0.00	0.00	0.00
4	*β*-Ocimene	38.51	0.02	0.00	0.00	0.03
9	Bornyl acetate	13.13	0.04	0.19	0.09	0.06
10	Undecanone	98.50	0.00	0.00	0.00	0.00
11	4-Terpineol	3.31	1.13	1.46	1.84	1.13
15	*α*-Terpineol	1.00	1.51	1.63	2.58	2.16
18	*β*-Citronellol	3.02	0.47	0.35	0.00	0.63

BET values were collected in the water matrix; odor activity values (OAV) were calculated for each compound using the concentration found in Table 1 and the orthonasal BET values in this study.

**Table 3 foods-09-00638-t003:** Descriptive analysis result of omija fruit extracts.

Aroma Attribute	Definition	Raw Omija Fruit Extract	Pureed Omija Fruit Extract	Freeze Dried Omija Fruit Extract	Hot Air-Dried Omija Fruit Extract	*p*-Value
Overall Aroma Intensity	Overall aroma impact	1.23 b	1.77 b	2.50 a	2.83 a	0.002
Cough syrup	Characteristic aromatics associated with cherry cough syrup (Reference: Bruffen^®^, Samil-pharm)	1.23 a	1.00 a	1.03 a	1.77 a	0.053
Spicy	Characteristic aromatics associated with whole black pepper (Reference: whole black pepper, Ottugi^®^)	0.53 b	0.83 b	1.43 a	1.90 a	0.002
Smoky aromatics	Characteristic aromatics associated with burning wood at campfire	0.70 a	0.17 b	0.17 b	0.67 a	0.033
Wet wood	Characteristic aromatics associated with wet wood	1.50 a	0.50 b	1.50 a	2.17 a	0.014
Lemon	Characteristic aromatics associated with cooked lemon (Reference: freshly shaved lemon peel)	0.90 a	0.60 a	0.73 a	0.67 a	0.606
Tomato ketchup	Characteristic aromatics associated with tomato ketchup (Reference: tomato ketchup, Ottugi^®^)	1.17 a	0.43 b	0.00 b	0.00 b	0.001
Alcohol	Characteristic aromatics associated with ethyl alcohol (Reference: 10% (*v*/*v*) ethyl alcohol in distilled water)	1.10 c	1.23 c	1.83 b	3.27 a	<0.0001

Numbers represent the mean value of aroma intensity of each sensory term, on a 15-pt universal scale using the Spectrum™ method; Means in a row that do not share the same alphabetical letter represent significant differences at *p* < 0.05.

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
