# Peer review of "Characterization of Key Aroma-Active Compounds Isolated from Omija Fruit Treated Differently Based on Odor Activity Values and Descriptive Sensory Analysis"

_foods, 2020, doi:10.3390/foods9050638_

Round 1

Reviewer 1 Report

Introduction provides all necessary information and facilitates the reader to develop understanding of Omija and recent development in that area.  Materials and methods are well and accurately described. The discussion is comprehensive and  helps in understating of the obtained data. Moreover, it provides valuable references to the literature. Conclusions are sufficient. The article could be published almost in present form, however I have just few remarks/suggestion.

In line 22 extracted with ethanol instead of in.

In line 38 Kimchi is placed two times (one as Kimchi and the other as kimchi with added omija)

In lines 81 and 89 use the same unit, I suggest Pascals.

In line 84-85 repetition of information provided in lines 77-78.

Information in line 161-162 is redundant for scientific value of the manuscript. Please consider moving that to acknowledgments?

Author Response

Reviewer 1

Introduction provides all necessary information and facilitates the reader to develop understanding of Omija and recent development in that area.  Materials and methods are well and accurately described. The discussion is comprehensive and helps in understating of the obtained data. Moreover, it provides valuable references to the literature. Conclusions are sufficient. The article could be published almost in present form, however I have just few remarks/suggestion.

In line 22 extracted with ethanol instead of in.

Answer: Sentence revised per reviewer’s request and we truly thank reviewer for the comment (L22)

In line 38 Kimchi is placed two times (one as Kimchi and the other as kimchi with added omija)

Answer: Sentence and reference number revised accordingly per reviewer’s request. (L39)

In lines 81 and 89 use the same unit, I suggest Pascals.

Answer: Units in L82 changed to Pa per reviewer’s request, and we truly thank reviewer for the comment

In line 84-85 repetition of information provided in lines 77-78.

Answer: Sentence in L 84-85 deleted per reviewer’s request

Information in line 161-162 is redundant for scientific value of the manuscript. Please consider moving that to acknowledgments?

Answer: Sentence removed as it is not relevant to the scientific value of the manuscript. We thank reviewer for the valuable comment (L162-163)

Reviewer 2 Report

This paper describes study aimed to characterize the key aroma-active compounds isolated from omija fruits. This goal was achieved by means chromatography–mass spectrometry (GC-MS), orthonasal threshold testing and descriptive sensory analysis techniques. Omija fruits were prepared using four different methods: raw, pureed, freeze dried and hot-air dried, and studied with help of instrumental analysis and sensory panels.

The major drawback of presented paper is the lack of consistency of presented data. This is especially evident when PCA results are compared with instrumental and sensory data. Moreover, the conclusions are not in line with the results.

The PCA bilplot presents data from instrumental and sensory analyses. There are some general rules for PCA biplot analysis:

  • if the angle between two variable vectors is small, they are strongly associated
  • the origin represents the average value for each variable, that is, it represents the object that has an average value in each variable
  • the larger the projection of an object on a variable vector, the more this object deviates from the average in the variable.
  • an object (variable) at a large distance from the origin has a large interaction effect with at least one variable (object);

1. Considering the above, based on presented PCA biplot it can be concluded that hot-air dried omija samples have the highest interaction with sensory and instrumental variables. The same goes to raw omija and pureed omija. The freeze-dried omija samples are at the closest proximity with the plot origin, which means rather weak interaction with variables. How authors justify this observation with statement from lines 302-303: “Freeze-dried omija fruits had the most versatile flavor profiles than other omija fruits treated differently (raw omija, pureed omija, and hot air-dried omija fruit).”

2. For hot-air dried samples 4-Terpieneol and α-Terpieneol were rated as highly odor active compounds (OVA equal to 1.13 and 2.16 respectively, Table 2), however the PCA biplot showed that impact of both compounds (indicated by similar angle between variable vectors) on hot-air dried samples is as strong as in case of γ-terpinen and 1,8-Cineol which OVA were rated as 0.0 and 0.2 respectively (Table 2). Could authors explain these discrepancies ? Why PCA analysis showed similar impact of variables which OVA values differed ten times ?

3.Based on fig.1 in lines 288-290 authors claim that “Hot-air dried omija were characterized with high intensities of 1,8-cineol, α-, γ-terpinene and α-, and 4-terpineol compounds with sensory characteristics of cough syrup, spicy and alcohol. 1,8-cineol are known as eycaliptol, with spice-like aromatics”. However on the same plot the freeze-dried omija is on the opposite side of the plot F1  axis than hot-air dried omija. Remembering that the larger the projection of an object on a variable vector, the more this object deviates from the average in the variable, and that object at a large distance from the origin has a large interaction effect with at least one variable, it can be concluded that freeze-dried omija has much weaker interaction with γ-terpinene and α-, and 4-terpineol compounds than hot-air dried omija. However, this was not confirmed by the data collected in tables 1 and 2 which showed the highest average values of OVA and amount of these compounds for freeze-dried omija. How author can explain these differences ?

4. Line 288: “…..of 1,8-cineol, α-, γ-terpinene and α-, and 4-terpineol……..” There is no  α-terpinene in this study, there is only γ-terpinene.

5. Table 3 – the treatments does not correspond with the rest of the manuscript, there is no “hot air-dried omija” in this table which is present in the rest of the manuscript and tables.

6.Lines 304-305: “This study found that 4-terpineol and α-terpineol are high odor-active compounds in omija fruits, with high OAVs and samples with high levels of these two compounds (freeze-dried omija fruit) had high intensities of “spicy” and “wet-wood” aromatics.” Based on the angle between variable vectors the 4-terpineol is indeed characterized by spicy taste but, α-terpineol showed equally similar correspondence with alcoholic and spicy tastes. The wet-wood aromatics were corresponding to b-Myrcene and γ-terpinene. This is in contrast to the claims from the conclusions.

7.Lines 294-295: “A higher correlation of undecanone and β-ocimene to tomato ketchup and lemon-like aromatics, and these were characteristics aromatics of raw omija fruits”. The OVA of undecanone for raw omija was rated as 0.0. How this is possible that this compound shows such high association with raw omija and its aroma profile, while having OVA equal to 0.0 ?

8. Despite the above doubts it is rather obvious that undecanone and β-ocimene will show some correlation with aroma of raw omija since both compounds were found only in these samples. However, this puts into question the correctness of fig. 1, because based on the object positions (points indicating samples) and variable vectors it can be concluded that there is some association between both compounds and the rest of the samples (pureed, freeze dried etc.). How this is possible since both compounds were not found in any other samples than raw omiija ? Shouldn’t PCA show no association at all or purely negative association (in other words, characterized by lack of these compounds) between both compounds and rest of the samples ?

Author Response

Reviewer 2

This paper describes study aimed to characterize the key aroma-active compounds isolated from omija fruits. This goal was achieved by means chromatography–mass spectrometry (GC-MS), orthonasal threshold testing and descriptive sensory analysis techniques. Omija fruits were prepared using four different methods: raw, pureed, freeze dried and hot-air dried, and studied with help of instrumental analysis and sensory panels.

The major drawback of presented paper is the lack of consistency of presented data. This is especially evident when PCA results are compared with instrumental and sensory data. Moreover, the conclusions are not in line with the results.

The PCA bilplot presents data from instrumental and sensory analyses. There are some general rules for PCA biplot analysis:

  • if the angle between two variable vectors is small, they are strongly associated
  • the origin represents the average value for each variable, that is, it represents the object that has an average value in each variable
  • the larger the projection of an object on a variable vector, the more this object deviates from the average in the variable.
  • an object (variable) at a large distance from the origin has a large interaction effect with at least one variable (object);

  1. Considering the above, based on presented PCA biplot it can be concluded that hot-air dried omija samples have the highest interaction with sensory and instrumental variables. The same goes to raw omija and pureed omija. The freeze-dried omija samples are at the closest proximity with the plot origin, which means rather weak interaction with variables. How authors justify this observation with statement from lines 302-303: “Freeze-dried omija fruits had the most versatile flavor profiles than other omija fruits treated differently (raw omija, pureed omija, and hot air-dried omija fruit).”

Answer: The statement was written based on the descriptive sensory analysis results, as can be seen in Table 3. Sentence in the conclusion (Freeze-dried omija fruits had the most versatile flavor profiles than other omija fruits…” revised to clarify the intended meaning (L319)

  1. For hot-air dried samples 4-Terpieneol and α-Terpieneol were rated as highly odor active compounds (OVA equal to 1.13 and 2.16 respectively, Table 2), however the PCA biplot showed that impact of both compounds (indicated by similar angle between variable vectors) on hot-air dried samples is as strong as in case of γ-terpinen and 1,8-Cineol which OVA were rated as 0.0 and 0.2 respectively (Table 2). Could authors explain these discrepancies ? Why PCA analysis showed similar impact of variables which OVA values differed ten times ?

Answer: We truly thank reviewer for the valuable comment. The reviewer pointed out the important issues with flavor analysis. OAV was calculated based on the concentration found in food matrix (such as data found Table 1) and sensory threshold value (such as data found in Table 2). The OAV demonstrated the high aroma impact compounds in food matrix, and the reason(s) why we included OAV in this study was to define key aroma active compounds in omija fruits prepared differently. PCA biplot is to demonstrate how each omija fruit is located in sensory and instrumental flavor analysis map, meaning sensory data and instrumental flavor analysis results were considered.

Based on the instrumental flavor analysis results (Table 1), concentration of γ-terpinen and 1,8-Cineol were higher than concentrations found in other samples. That is the reason(s) why high correlations of two compounds in hot-air dried omija fruits were observed in PCA biplot (Figure 1). While PCA biplot and instrumental flavor analysis results demonstrated that these two compounds (γ-terpinen and 1,8-Cineol) were highly associated with hot-air dried omija samples, OAV results explained that these two compounds may not be the key aroma active compounds in hot-air dried omija sample. As stated earlier, OAV considered not only concentration of compounds found in food, but also considers orthonasal sensory threshold perceived by human. Therefore, OAV fills the gap between peak area found in GC-MS and actual human perception.

To answer the question posed by reviewer regarding the discrepancy between PCA and OAV values of γ-terpinen and 1,8-Cineol in hot air-dried omija sample, PCA biplot was drawn based on instrumental flavor analysis results (Table 1) and sensory data. PCA biplot reflects the concentration found in omija samples. OAV, however, explains the gap between instrumental analysis results and human sensory perception. Therefore, the discrepencies were as expected. The explanation regarding the discrepancies were included in the manuscript in L292-298, because it is worth to note in the manuscript.

3.Based on fig.1 in lines 288-290 authors claim that “Hot-air dried omija were characterized with high intensities of 1,8-cineol, α-, γ-terpinene and α-, and 4-terpineol compounds with sensory characteristics of cough syrup, spicy and alcohol. 1,8-cineol are known as eycaliptol, with spice-like aromatics”. However on the same plot the freeze-dried omija is on the opposite side of the plot F1  axis than hot-air dried omija. Remembering that the larger the projection of an object on a variable vector, the more this object deviates from the average in the variable, and that object at a large distance from the origin has a large interaction effect with at least one variable, it can be concluded that freeze-dried omija has much weaker interaction with γ-terpinene and α-, and 4-terpineol compounds than hot-air dried omija. However, this was not confirmed by the data collected in tables 1 and 2 which showed the highest average values of OVA and amount of these compounds for freeze-dried omija. How author can explain these differences ?
Answer: We again, thank reviewer for pointing out the important issues in PCA biplot. Please note that PCA biplot is 2-dimensional demonstration of 3-dimensional data space. This PCA biplot in Figure 1 explains 85.42% of total variability of dataset, meaning that 14.58% of data is not explained by PCA biplot. These missing data is in F3 and F4 panel. When checking on F3 and F4 panel, explaining 9.4 and 5.18% of total variability, α-, and 4-terpineol were highly impact compounds in F3 panel, which was not utilized in PCA biplot in this study. (We cross-checked F3 and F4 panel to confirm this statement). Multivariate statistical analysis such as PCA may miss out some of the dataset, and this is the case of missing data. While some details are missing, current PCA still explains ~85% of total variability and correlation between volatile compounds and sensory aroma perception is still well-explained, therefore I would keep PCA biplot as it is. We included the statement regarding the limitation of PCA biplot in L286-288

  1. Line 288: “…..of 1,8-cineol, α-, γ-terpinene and α-, and 4-terpineol……..” There is no  α-terpinene in this study, there is only γ-terpinene.
    Answer: We thank reviewer for the comment, and “α-terpinene” is deleted in the statement. (L293)
  2. Table 3 – the treatments does not correspond with the rest of the manuscript, there is no “hot air-dried omija” in this table which is present in the rest of the manuscript and tables.
    Answer: We checked the treatment names in Table 3 and revised Table 3 and the rest of the manuscript accordingly.

6.Lines 304-305: “This study found that 4-terpineol and α-terpineol are high odor-active compounds in omija fruits, with high OAVs and samples with high levels of these two compounds (freeze-dried omija fruit) had high intensities of “spicy” and “wet-wood” aromatics.” Based on the angle between variable vectors the 4-terpineol is indeed characterized by spicy taste but, α-terpineol showed equally similar correspondence with alcoholic and spicy tastes. The wet-wood aromatics were corresponding to b-Myrcene and γ-terpinene. This is in contrast to the claims from the conclusions.

Answer: Whole section in PCA biplot (L282-312) as well as conclusion (L314-324) majorly revised according to reviewer’s valuable comments. We hope this now be suitable for the publication in Foods. 

7.Lines 294-295: “A higher correlation of undecanone and β-ocimene to tomato ketchup and lemon-like aromatics, and these were characteristics aromatics of raw omija fruits”. The OVA of undecanone for raw omija was rated as 0.0. How this is possible that this compound shows such high association with raw omija and its aroma profile, while having OVA equal to 0.0 ?

Answer: Statement regarding undecanone and β-ocimene is revised accordingly as well as whole section explaining PCA biplot as well as conclusion (L282-324) majorly revised according to reviewer’s valuable comments. We hope this now be suitable for the publication in Foods. 

  1. Despite the above doubts it is rather obvious that undecanone and β-ocimene will show some correlation with aroma of raw omija since both compounds were found only in these samples. However, this puts into question the correctness of fig. 1, because based on the object positions (points indicating samples) and variable vectors it can be concluded that there is some association between both compounds and the rest of the samples (pureed, freeze dried etc.). How this is possible since both compounds were not found in any other samples than raw omiija ? Shouldn’t PCA show no association at all or purely negative association (in other words, characterized by lack of these compounds) between both compounds and rest of the samples ?

Answer: We truly value the reviewer’s comments and thank reviewer for the feedback. Please note that PCA biplot in this study still does not explain ~15% of total variability in the total dataset. I do not think PCA can show the “no association” in current 2D space. We checked the correlation matrix to confirm the correctness of Fig 1. The section regarding PCA biplot is majorly revised based on the valuable comments mentioned in above, so this now be suitable for the publication in Foods.

Reviewer 3 Report

In my opinion the paper is well written and scientifically sound. Thus, it is suitable for publication in the present form.

Author Response

Reviewer 3

In my opinion the paper is well written and scientifically sound. Thus, it is suitable for publication in the present form.

Answer: We truly thank reviewer for valuable comment.

Round 2

Reviewer 2 Report

Authors comment

We again, thank reviewer for pointing out the important issues in PCA biplot. Please note that PCA biplot is 2-dimensional demonstration of 3-dimensional data space. This PCA biplot in Figure 1 explains 85.42% of total variability of dataset, meaning that 14.58% of data is not explained by PCA biplot. These missing data is in F3 and F4 panel. When checking on F3 and F4 panel, explaining 9.4 and 5.18% of total variability, α-, and 4-terpineol were highly impact compounds in F3 panel, which was not utilized in PCA biplot in this study. (We cross-checked F3 and F4 panel to confirm this statement). Multivariate statistical analysis such as PCA may miss out some of the dataset, and this is the case of missing data. While some details are missing, current PCA still explains ~85% of total variability and correlation between volatile compounds and sensory aroma perception is still well-explained, therefore I would keep PCA biplot as it is. We included the statement regarding the limitation of PCA biplot in L286-288 

Reviewers comment

I have to completely disagree with this statement. There is no limitation in presentation of data using biplots, only the limited data is presented in this study. The PCA analysis does not miss the data, especially if data is relevant. PCA is a orthogonal linear transformation of variable data space to new coordinate system, which maximize the impact of relevant variables. Principal components are the axes of new coordinate system, which are orthogonally oriented to each other. Therefore the pca biplot is simply a projection of data on a plane created by two axes. Any two axes can be selected to project the data set and in most cases it is a trivial operation. Authors state that “α-, and 4-terpineol were highly impact compounds in F3 panel, which was not utilized in PCA biplot in this study”. Since authors claim that 14.58% of data variability was explained by third and fourth components of PCA, and that this portion of data was highly relevant, I strongly insist to include the F3-F4 or F2-F3 biplots in the manuscript. Please show this data to support claimed impact of α-, and 4-terpineol compounds.

Authors comment

We truly value the reviewer’s comments and thank reviewer for the feedback. Please note that PCA biplot in this study still does not explain ~15% of total variability in the total dataset. I do not think PCA can show the “no association” in current 2D space. We checked the correlation matrix to confirm the correctness of Fig 1. The section regarding PCA biplot is majorly revised based on the valuable comments mentioned in above, so this now be suitable for the publication in Foods.

Reviewers comment

I do not expect the PCA to show no association by itself. My point was that the current associations can by highly biased by the data selection. Currently data shows some association between compounds that are not present at all in some of the samples. Maybe it would be better to conduct PCA using only compounds that were detected in all samples ? And again, PCA is not limited to 2D space. It is a multidimensional technique that can be visualized using 2D projections of any considered dimension.  

Minor issues:

Please improve the quality of data presentation in tables 1 and 3. For instance adjust the width of KI column in table 1 to show the data below in one line (now data is split into two lines which is unreadable) . In table 3 it is difficult to link aroma attributes and their definitions. Please add some line spacing to properly link the definitions with the attributes.

Lines 281-285: Authors describe pca axes as "PC1" and "PC2", however on biplot use "F1" and "F2". Please improve this issue.

Author Response

Authors comment

We again, thank reviewer for pointing out the important issues in PCA biplot. Please note that PCA biplot is 2-dimensional demonstration of 3-dimensional data space. This PCA biplot in Figure 1 explains 85.42% of total variability of dataset, meaning that 14.58% of data is not explained by PCA biplot. These missing data is in F3 and F4 panel. When checking on F3 and F4 panel, explaining 9.4 and 5.18% of total variability, α-, and 4-terpineol were highly impact compounds in F3 panel, which was not utilized in PCA biplot in this study. (We cross-checked F3 and F4 panel to confirm this statement). Multivariate statistical analysis such as PCA may miss out some of the dataset, and this is the case of missing data. While some details are missing, current PCA still explains ~85% of total variability and correlation between volatile compounds and sensory aroma perception is still well-explained, therefore I would keep PCA biplot as it is. We included the statement regarding the limitation of PCA biplot in L286-288 

Reviewers comment

I have to completely disagree with this statement. There is no limitation in presentation of data using biplots, only the limited data is presented in this study. The PCA analysis does not miss the data, especially if data is relevant. PCA is a orthogonal linear transformation of variable data space to new coordinate system, which maximize the impact of relevant variables. Principal components are the axes of new coordinate system, which are orthogonally oriented to each other. Therefore the pca biplot is simply a projection of data on a plane created by two axes. Any two axes can be selected to project the data set and in most cases it is a trivial operation. Authors state that “α-, and 4-terpineol were highly impact compounds in F3 panel, which was not utilized in PCA biplot in this study”. Since authors claim that 14.58% of data variability was explained by third and fourth components of PCA, and that this portion of data was highly relevant, I strongly insist to include the F3-F4 or F2-F3 biplots in the manuscript. Please show this data to support claimed impact of α-, and 4-terpineol compounds.

Answer: Whole section regarding PCA biplot is re-written based on valuable comments from reviewer. We truly thank reviewer for giving valuable insight on PCA and we tried our best to interpret the PCA biplots according to the valuable comment given by reviewer. PCA biplot with PC1 and PC2 (figure 1a) and PCA biplot with PC1 and PC3 (figure 1b) are now added and interpretation on each PCA biplots were added in the manuscript. The interpretation of α-, and 4-terpineol as aroma active compounds in omija fruit extracts were revised accordingly. We hope this now be acceptable as  proper interpretation of the dataset. Please find L286-360 for newly written section regarding interpretation of PCA biplots.

Authors comment

We truly value the reviewer’s comments and thank reviewer for the feedback. Please note that PCA biplot in this study still does not explain ~15% of total variability in the total dataset. I do not think PCA can show the “no association” in current 2D space. We checked the correlation matrix to confirm the correctness of Fig 1. The section regarding PCA biplot is majorly revised based on the valuable comments mentioned in above, so this now be suitable for the publication in Foods.

Reviewers comment

I do not expect the PCA to show no association by itself. My point was that the current associations can by highly biased by the data selection. Currently data shows some association between compounds that are not present at all in some of the samples. Maybe it would be better to conduct PCA using only compounds that were detected in all samples ? And again, PCA is not limited to 2D space. It is a multidimensional technique that can be visualized using 2D projections of any considered dimension.  

Answer: I acknowledge the false understanding of PCA biplot, and we again thank reviewer for giving valuble insights on PCA. We now revised the whole section on PCA biplots according to the valuable comment given by reviewer (L286-360), and tried our best not to falsely interpret the data. Regarding the compound selection for PCA biplot, we selected 9 compounds for PCA because these were the previously reported compounds that are present in omija fruits in higher concentration and were known odor-active compounds based on the literature reviews. We thought about including only compounds that were detected in all samples for PCA before. With many thoughts and consideration based on current literature reviews, we decided to include these 9 compounds. These 9 compounds were the compounds included for OAV calculation in this study (Table 2) as well. We hope that selection of 9 compounds for PCA in this study is acceptable by reviewer.

Minor issues:

Please improve the quality of data presentation in tables 1 and 3. For instance adjust the width of KI column in table 1 to show the data below in one line (now data is split into two lines which is unreadable) . In table 3 it is difficult to link aroma attributes and their definitions. Please add some line spacing to properly link the definitions with the attributes.

Lines 281-285: Authors describe pca axes as "PC1" and "PC2", however on biplot use "F1" and "F2". Please improve this issue.

Answer: Minor issues pointed out by reviewer are all revised accordingly. For example, Tables 1 and 3 are revised to clearly visualize the data by adjusting the cell width (Table 1), and line spacing (Table 3). Issues with F1 and F2 in PCA biplots are now revised accordingly.